Home range size and habitat selection of owned outdoor domestic cats (Felis catus) in urban southwestern Ontario

Pyott Marlee L. pyottmarlee@gmail.com 1
Norris D. Ryan 1
Mitchell Greg W. 2 3
Custode Leonardo 1
Gow Elizabeth A. 1 4
1 Integrative Biology, University of Guelph , Guelph , Ontario , Canada
2 Department of Integrative Biology, Carleton University , Ottawa , Ontario , Canada
3 Wildlife Research Division, Environment and Climate Change Canada, National Wildlife Research Centre , Ottawa , Ontario , Canada
4 Wildlife Research Division, Environment and Climate Change Canada, Pacific Wildlife Research Centre , Delta , British Columbia , Canada
Vonk Jennifer
Electronic publication date: 2024 Mar 29
Publication date: 2024
Volume: 12
Electronic Location ID: e17159
Received 2023 Nov 28; Accepted 2024 Mar 4
Copyright: ©2024 Pyott et al.
Copyright year: 2024
Copyright holder: Pyott et al.
License: This is an open access article distributed under the terms of the Creative Commons Attribution License, which permits unrestricted use, distribution, reproduction and adaptation in any medium and for any purpose provided that it is properly attributed. For attribution, the original author(s), title, publication source (PeerJ) and either DOI or URL of the article must be cited.
License URL: https://creativecommons.org/licenses/by/4.0/

Keywords: Cat management, Free-ranging cat, Free-roaming cat, GPS, Home range size, Habitat selection, Kernel density estimate, Minimum convex polygon

Funding: The Kenneth Molson Foundation Environment and Climate Change Canada The Liber Ero Foundation The Weston Family Foundation through the Nature Conservancy of Canada The Natural Sciences and Engineering Research Council of Canada, and an Ontario Graduate Scholarship Funding for this project was provided by the Kenneth Molson Foundation, Environment and Climate Change Canada, the Liber Ero Foundation, the Weston Family Foundation through the Nature Conservancy of Canada, the Natural Sciences and Engineering Research Council of Canada, and an Ontario Graduate Scholarship. The funders had no role in study design, data collection and analysis, decision to publish, or preparation of the manuscript.

==============================
Domestic cats (Felis catus) play a dual role in society as both companion animals and predators. When provided with unsupervised outdoor access, cats can negatively impact native wildlife and create public health and animal welfare challenges. The effective implementation of management strategies, such as buffer zones or curfews, requires an understanding of home range size, the factors that influence their movement, and the types of habitats they use. Here, we used a community/citizen scientist approach to collect movement and habitat use data using GPS collars on owned outdoor cats in the Kitchener-Waterloo-Cambridge-Guelph region, southwestern Ontario, Canada. Mean (± SD) 100% minimum convex polygon home range size was 8 ± 8 ha (range: 0.34–38 ha) and was positively associated with road density but not with intrinsic factors such as boldness, sex, or age. With regards to habitat selection, cats used greenspaces, roads, and agricultural land less often than predicted but strongly selected for impervious surfaces (urban areas other than greenspaces or roads). Our results suggest that wildlife near buildings and residential areas are likely at the greatest risk of cat predation and that a buffer size of 840 m would be needed to restrict cats from entering areas of conservation concern.

Introduction

Domestic cats (Felis catus) are the second most popular pet in the world (Growth from Knowledge, 2016) and are often at the centre of controversy because of the dual role they play in society as both companion animals and wild predators (Crowley, Cecchetti & McDonald, 2020a). Their retained keen predatory senses are reflective of their shared history alongside humans as useful agricultural pest controllers (Fitzgerald & Turner, 2000; Driscoll, Macdonald & O’Brien, 2009; Montague et al., 2014). However, retention of predatory traits also means that cats can impact native wildlife through hunting. Globally, cats are estimated to kill billions of wild animals around the world each year (Woods, McDonald & Harris, 2003; Blancher, 2013; Loss, Will & Marra, 2013; Woinarski et al., 2017; Mori et al., 2019; Seymour et al., 2020). In Canada alone, after habitat destruction and degradation, cats were estimated to be the largest contributor of direct bird mortality (Calvert et al., 2013), estimated to kill between 100–350 million birds annually (Blancher, 2013).

Unsupervised outdoor cats can also create public health and animal welfare challenges. Cats can carry a wide variety of pathogens, including Toxoplasma gondii (Flegr et al., 2014; Hamilton et al., 2015), cat scratch disease Bartonella henselae (Kravetz & Federman, 2002), and feline leukemia (Brown et al., 2008; Bevins et al., 2012). Many of the zoonoses carried by cats have the potential to spread to humans, wildlife, and livestock. Outdoor cats can also be seriously injured or killed by ingesting harmful substances (Heyward & Norbury, 1999; Fitzgerald, 2010), being attacked by coyotes Canis latrans (Grubbs & Krausman, 2009) or hit by vehicles (Olsen & Allen, 2001). Negative consequences that a cat might face while outside can also impact an owner’s welfare or health since their well-being is often intertwined with their pet (Amiot & Bastian, 2015; Finka et al., 2019).

In several ways, urban environments can magnify the impacts and risks of providing cats with outdoor access when compared to rural areas. Urbanization has dramatically altered landscapes by replacing vegetation with impervious surfaces leading to a loss of natural habitats and biodiversity (Alberti et al., 2020; Simkin et al., 2022). These changes in land cover and land use have forced humans and wildlife to live closer together, intensifying human-wildlife interactions and conflict (Soulsbury & White, 2015). In addition, because the abundance of cats is positively associated with human density (Sims et al., 2007; Flockhart, Norris & Coe, 2016; Hand, 2019), their presence in urban areas may concentrate predation pressure (Thomas, Fellowes & Baker, 2012; Hanmer, Thomas & Fellowes, 2017; Kays et al., 2020) on native wildlife, especially for those species that thrive in urban ecosystems, as well as amplify the transmission risk for zoonotic disease (Mackenstedt, Jenkins & Romig, 2015; Bolais et al., 2017; Candela et al., 2022). There is also a concern for the transmission of cat-related zoonotic disease disproportionately affecting low-income communities which, in some cities, have the highest density of cats (Flockhart, Norris & Coe, 2016; McDonald & Clements, 2019).

Collaborative approaches for managing owned outdoor cats have often been geared towards recognizing the concerns and needs of cat owners by not banning cats from being outside completely. Instead, management strategies have focused on imposing restrictions on areas or time periods when cats can be outside, such as implementing policies that enforce nightly cat curfews (Grayson & Calver, 2004; Legge et al., 2020; Thomas, Fellowes & Baker, 2012) or introducing buffer zones around natural areas i.e., park, game reserve, or protected area; (Heinen & Mehta, 2000; Lilith, Calver & Garkaklis, 2008; Pirie, Thomas & Fellowes, 2022). Buffer zones may reduce the impact of outdoor cats on wildlife by forbidding residents from owning an outdoor cat within a set boundary (Lilith, Calver & Garkaklis, 2008). Previous research has suggested a variety of buffer zone sizes ranging from 300–2400 m, based on the maximum linear distance a cat travels from their owner’s house (Lilith, Calver & Garkaklis, 2008; Mesters, Seddon & van Heezik, 2010; Thomas, Baker & Fellowes, 2014; Hanmer, Thomas & Fellowes, 2017), and cat home range size (Mesters, Seddon & van Heezik, 2010; Thomas, Baker & Fellowes, 2014; Herrera et al., 2022). This variation in buffer zone recommendations suggests that movement may be dependent on a variety of factors, including the composition of the landscape, habitats within a study area, or even the methods used to track individuals.

Several factors may influence the size of an outdoor cat’s home range in urban areas. Previous research suggests that home ranges are larger for younger cats than older cats (Hall et al., 2016; Kays et al., 2020; Jensen et al., 2022), and for cats living in rural areas compared to urban areas (Hall et al., 2016; White, 2019; Bachmann, 2020). In addition, while some studies did not find evidence for differences between sexes (Meek, 2003; Mesters, Seddon & van Heezik, 2010; van Heezik et al., 2010; Thomas, Baker & Fellowes, 2014), others found that males had significantly larger home ranges than females (Hall et al., 2016; Roetman et al., 2017; Kays et al., 2020; Pirie, Thomas & Fellowes, 2022). Geographic location can also influence a cat’s home range size, as Kays et al. (2020) found that cats living in Australia tended to have smaller home ranges than cats living in the United Kingdom, U.S.A., and New Zealand, speaking to the need for more studies across a range of cities and geographies.

In addition to the size of a cat’s home range, understanding how individuals select or avoid certain habitats may also be important for effective management. Habitat selection has been described as a hierarchical process, with establishment of a species’ geographical range (first-order selection) and then, within that space, individuals choose a home range (second-order selection) where they travel in search of resources (Burt, 1943; Johnson, 1980). Within their home range, individuals show fine-scale movements that pertain to more specific habitats (third-order selection; Johnson, 1980). First and second order selection does not apply to owned cats because their range distribution (first-order) and location of their home range (second-order) are determined by owners. However, owned cats with unrestricted access to the outdoors may show selection for specific habitat types within their home range. While prior studies have reported that cats spend the majority of their time in natural greenspaces (Barratt, 1997; Meek, 2003; van Heezik et al., 2010; Thomas, Baker & Fellowes, 2014), others have provided evidence that cats spend most of their time in disturbed habitat, close to buildings (Mesters, Seddon & van Heezik, 2010; Kays et al., 2020; Fardell et al., 2021; Bischof et al., 2022). These differences in habitat selection of cats may be due to variation in predator communities, climate, or other landscape features, which may differ in geographic location, or differing methodology emphasizing why more research is needed to help better inform specific cat management strategies.

Determining patterns of movement and habitat selection of owned outdoor cats can provide valuable information for a variety of stakeholders, such as cat owners, conservationists, shelter workers, veterinarians, and policymakers. The size of a cat’s home range can provide a buffer zone size based on the best available science (Lilith, Calver & Garkaklis, 2008; Thomas, Baker & Fellowes, 2014) and determining how far cats tend to roam can help predict transmission of pathogens between cats, livestock, wildlife, and humans, or where disease hotspots may occur (Han, Kramer & Drake, 2016). The types of habitats cats select can help identify potential welfare risks and the magnitude of the risks for outdoor cats, which can help shelter workers and veterinarians advise cat owners on the costs and benefits of providing unsupervised outdoor activity. For example, if cats in urban areas tend to walk down roads, they may be at an elevated risk of being hit by a vehicle. If outdoor cats are spending a large portion of their time in greenspaces this may increase the risk of cats encountering coyotes, a known predator of cats (Quinn, 1997; Morey, Gese & Gehrt, 2007; Grubbs & Krausman, 2009). Additionally, determining habitat selection of outdoor cats can help stakeholders understand if cats use fragmented habitat patches that may lead to higher rates of contact with wildlife (Ives et al., 2016; Threlfall et al., 2017).

Here, we used a community/citizen-scientist (Dickinson & Bonney, 2012) approach in which owners living in the Kitchener-Waterloo-Cambridge-Guelph region of southwestern Ontario, Canada, worked with us to collect movement data from their cat with unsupervised outdoor access using Global Positioning System (GPS) technology. Our objectives were to estimate a cat’s home range size, identify what intrinsic and habitat related factors may influence their home range size, and quantify the land cover types used and selected for by cats. Specifically, we examined the following hypotheses: there would be larger home ranges (1) for male cats because cats are polygamous so males would benefit from maintaining larger home ranges than females, in theory, to access multiple mating opportunities (Liberg et al., 2000; Palomares et al., 2017), (2) during night because cats are nocturnal (Kuwabara, Seki & Aoki, 1986; Barratt, 1997), (3) in relatively younger cats because elderly cats are more lethargic and could have other age-related complications that impact their ability to move (Hall et al., 2016; Kays et al., 2020), (4) in cats with fewer roads located near their home because road traffic can act as a barrier for movement (Barratt, 1997), and (5) in cats with a bold personality (a low neurotic score) because they are willing to roam farther than cats that are shy (high neurotic score; Litchfield et al., 2017). Finally, at the habitat selection level, we examined the hypotheses that cats avoid roads and greenspaces because of the risk associated with (6) traffic (Barratt, 1997) and (7) coyotes (Gehrt et al., 2013; Clyde et al., 2022).

Materials & Methods

Study Site

Movement data on owned cats with unsupervised outdoor access were collected from Aug.–Nov. 2019 and May–Oct. 2021 in the Kitchener-Waterloo-Cambridge-Guelph region of southwestern Ontario (Fig. 1). The cities within this region contain over 1000 ha of parklands and have small urban cores, with a combined human population of 816,873 (Statistics Canada, 2021a; Statistics Canada, 2021b). Land cover in the region is made up of agricultural land, commercial land, natural open spaces, and mixed residential houses with most of the population residing in single-detached houses (Statistics Canada, 2021a; Statistics Canada, 2021b). Proportions of this text were previously published as part of a preprint (Pyott, 2023).

Figure 1 The study site showing the habitat types used in our analysis and the residential locations of owned cats who participated.

Map of owned cats who participated in our study (white pin, n = 42) in the Waterloo-Kitchener-Cambridge-Guelph region of southwestern Ontario, Canada (outlined in red on the inset map). Also shown on the map are habitat types used in the habitat selection analysis, obtained from the SOLRIS v.3 (Ontario Ministry of Natural Resources and Forestry, 2019b). This map was produced in ArcGIS Pro v.2.8 and projected into Universal Transverse Mercator (UTM) zone 17N, with the 1984 World Geodetic System (WGS).

Equipment

As part of a larger study, we collected location and video data from outdoor cats by attaching a GPS and camera to a cat break-way collar. For the purpose of this study, we solely focused on the GPS data. The collars (GPS, 35 g; Catcam, 70 g; collar, 9 g), which weighed less than ∼5% of the smallest cat’s body weight (range: 2.3–8.6 kg), had an animal-borne camera on the front that rested below the cat’s chin, and a GPS unit on the back/top of the collar, that rested on the neck (Fig. 2; Tractive models TRATR3G and TRNJA4; Pasching, Austria). Owners were provided with the equipment for 5 wks and asked to collect at least 20 d of their cat’s outdoor activity. We asked owners to collect 20 d of data within a 5 wk period to increase the likelihood of compliance with the protocol and meet human ethics requirements by ensuring study participants had flexibility and did not feel forced to participate everyday. Despite the GPS having a standard battery life of 2–4 d, we instructed owners to charge the equipment for ∼2 hrs/d. The GPS units provided a fix of the cat’s location every 2–60 min, depending on the activity level of the cat and, in ideal conditions, had a positional accuracy of 8 m based on manufacturer details (Tractive, 2021) but with field testing accuracy was 4 ± 10 m in partial shade and 4.8 ± 1 m in areas fully exposed to the sky. All methods were approved by the University of Guelph’s Research Ethics Board (approval #4189), and the University of Guelph’s Animal Care Committee (AUP #4183).

Figure 2 The Catcam device used to track individual movements of owned outdoor cats.

(A) one of the cats that participated in our study, with the Catcam resting just below his chin. (B) the collar with the Catcam camera on the front and the GPS (indicated by an arrow) on the back.

Participants

We recruited cat owners by advertising through email listservs, social media, and news reports via the popular press and news stations. In our first year of data collection (2019), we enlisted 10 cats, and in our second year (2021), we enlisted 32 cats. In 2021, we added an online questionnaire (Article S1) that allowed us to record each cat’s sex, spay/neuter status, age, and personality score. The survey also allowed us to select participants with owned cats that spent unsupervised time outside (i.e., not in an enclosed outdoor space, or on a leash or tether) during both the day and night. Cats under 1 yr did not qualify because they were not fully grown, and their personalities could still be developing (Lowe & Bradshaw, 2001). Because the GPS units were attached to a collar and needed to be turned on by the owner before going outside (see details below), we also selected cats that had restricted outdoor access (i.e., did not use a cat door or other method of entering/exiting the house freely). All cats were selected from different owners and were classified as “owned indoor-outdoor” cats defined by (Crowley, Cecchetti & McDonald, 2019). Owned indoor-outdoor cats differ from free-ranging (e.g., cats that live outside entirely but have a colony caretaker that provides some shelter and food) and feral cats (no human influence) in that they have a central home they return to where they are fed and provided with shelter and owners have some control over cat movement (i.e., when a cat is put out) and reproduction (i.e., if a cat is fixed).

Owners were trained through written and video tutorials that we developed, including detailed instructions for attaching and operating the equipment. In 2019, meetings with owners occurred in-person, involving a single meeting for reviewing the informed consent form and teaching the owners how to use the equipment and fit the collar on their cat. Due to COVID-19 public health and University regulations restricting person-to-person contact, in 2021–22, we met with owners virtually to first go over the consent form and then again to observe them use the equipment and fit the collar around their cat. Before letting the cat outside with the equipment, owners provided their cat with a 1–2 d acclimation period where their cat wore the collar in the house so owners could observe if any abnormal behaviours occurred, such as excessive scratching at the collar, refusing to eat or play, or lethargy. If a cat showed any abnormal behaviour as reported by the owner, we removed them from the study. While owners had the equipment, we provided weekly check-up emails and video calls for technical assistance when required. As an incentive to participate, owners were given a map of their cat’s GPS tracks, a ‘highlight’ reel of their cat’s outdoor activity, a certificate of their cat’s personality score, and were entered into a draw for 1 of 4 $50 gift cards to a local pet store.

In total, we collected 63,474 GPS locations from 42 cats (18 males, 15 females, 9 unknown). Our sample size was limited by the number of camera devices, so no priori sample size calculations were performed. While we did not collect personal information about cats in 2019, we know through correspondence with owners that one was a male, and all were either neutered or spayed (hereafter, “desexed”). In 2021, out of 340 participants who responded to our survey, 121 owners had cats that qualified, and 34 were randomly selected to participate. One cat was withdrawn because the cat did not tolerate wearing the GPS, and one cat lost the tracker. This left 32 individuals (17 males, 15 females, all desexed), with the oldest cat being 14 yrs, and the youngest being 2.5 yrs (mean = 7 yrs, median = 6 yrs).

Filtering GPS data

We differentiated between “true” or “false” locations by filtering out points that were likely errors caused by interference between the GPS trackers and transmitting satellites (Moen, Pastor & Cohen, 1997; D’Eon et al., 2002). Specifically, to filter GPS data, we first determined what speed the cat would have traveled to reach each location based on the time and distance recorded between locations. Once we calculated a speed value for each GPS point in R v.4.1.2 (R Core Team, Vienna, Austria; Article S3), data were uploaded into ArcGIS Pro v.2.8 (Geographic Information System; ESRI, Redlands, CA., USA) for filtering. In ArcGIS, we removed highly improbable points based on date and time stamps (e.g., points that occurred at a researcher’s house or the University of Guelph campus). We then filtered out points over a speed threshold of 100 m/min (Recio & Seddon, 2013), classified as a trot (Smith, Chung & Zernicke, 1993), which was unlikely to be maintained by a cat for longer than 2 min. (Smith, Chung & Zernicke, 1993; Kim et al., 2014). Following this, we removed “spike locations”, which were characterized as points that could only occur in the unlikely event that a cat quickly ran to one location, then made a sharp turn to quickly return near the original location (Bjørneraas et al., 2010; Recio & Seddon, 2013). We used the same speed angle threshold values set by Recio & Seddon (2013) of 15 m/min with an outer turning angle between 165–180°. We then removed GPS points that occurred in-between points previously removed based on the above criteria. Finally, to further reduce bias in the movement and home range analyses, we removed all “stationary” points (points within 0 m of each other over successive locations) except the first point in a stationary series. We chose to do this because most stationary points occurred inside the owner’s home as a result of the owner not turning off the GPS immediately after the cat entered the house. Some stationary points may have been in the owner’s yard rather than the house, but we could not always distinguish these points from inside the house, so we chose the conservative approach to remove all stationary points. After filtering, there were 41,092 GPS points.

Home range size

We estimated home range size by using 100% minimum convex polygons (MCP) and 95% kernel density estimation (KDE). We analyzed both measures of home-range size to allow for comparison to other studies. Core home ranges, or 50% kernel density estimates (Samuel, Pierce & Garton, 1985; Heikkilä et al., 1996), are the most concentrated area of an individual’s activity (Burt, 1943) and exclude movement that occurs on the outskirts of the home range. Therefore, to estimate how far cats typically range, we focused on the 95% KDE and 100% MCP home ranges. To determine whether there were differences between the nocturnal and diurnal home range of cats, we split GPS points into day (06:00–17:59) and night (18:00–05:59). Because cats had sporadic schedules based on their owner’s behaviours and lifestyles, they would go in and out of the house relatively regularly when owners put them out. We based the specific timing of this split on when cats were inside for long periods of time. In the early morning, owners brought their cat inside to be fed before work, let them out again, brought them back in for a bit at the end of the workday, and put them out again sometime in the evening until the morning. These patterns were determined through conversations with owners. All home range analyses were carried out in R using the package ‘adehabitatHR’ (Calenge, 2006). We determined if sufficient GPS data were collected to accurately estimate home ranges by plotting the home range size vs. the number of GPS points. Through visual inspection for an asymptote (Harris et al., 1990), we determined that enough GPS points were collected for each cat (Article S4, Fig. S1).

Cat personality

To determine if a bold personality could impact home range size, we estimated the degree of neuroticism and assumed the inverse of this score was representative of boldness. Neurotic cats show traits of being shy and anxious, while cats that score low in neuroticism are the opposite —bold and secure (Litchfield et al., 2017). Neuroticism was measured by asking owners to complete a questionnaire (Article S5) based on an established personality test for domestic cats, called the ‘Feline Five’ (Litchfield et al., 2017). The Feline Five identified five personality types based on clusters of personality traits: (1) extraverted- traits such as inventive, persevering, and inquisitive; (2) agreeable- traits such as gentle, playful, affectionate, and friendly to people); (3) dominant- traits such as defiant, greedy, and unfriendly to other cats); (4) impulsive- traits such as distractible, erratic, and unpredictable; and (5) neurotic- traits such as shy, anxious, and insecure. The inter-item reliability of our personality measure was assessed using Cronbach’s alpha in R with the psych package (Revelle, 2024) and resulted in a coefficient of 0.92. Our questionnaire asked owners to evaluate 43 personality traits by rating how much they agreed or disagreed that their cat showed each trait on a 7-point Likert scale (Likert, 1932), with strongly disagree being the lowest score (1) and strongly agree as the highest score (7). Response scores were then summed and divided by the number of traits related to their respective personality type. Neuroticism was evaluated from the response scores of 13 traits (Litchfield et al., 2017) and we reported the standardized average score.

Road density

We estimated how road density and the presence or absence of major roads could impact home range size in ArcGIS by overlaying our GPS points of cat movement with road cover data from the Ontario Road Network (Ontario Ministry of Natural Resources and Forestry, 2019a). To estimate road density near each cat, we summed the length of roads sampled within a circular boundary centered on each individual’s mean coordinates. Because we wanted a boundary that captured how much road was available to each cat in a standardized way across cats, we used a radius of 66 m, which was based on an area the size of the median 95% KDE home range size. We examined if traffic acted as a barrier for cat movement by identifying roads as either “minor” or “major” based on Google Maps’ classification, which refers to traffic rates. We also “ground-truthed” (Wake & Hull, 1978) this classification by visiting the areas where all cats were tracked.

Land cover data

Land cover data were obtained from the Southern Ontario Land Resource Information System (SOLRIS) v.3 (resolution of +/−10 m and 30 m pixel resolution; Ontario Ministry of Natural Resources and Forestry, 2019b). We overlaid the land cover data with GPS locations from the cats that we tracked in ArcGIS. For our mapping purposes, we used the unmanipulated SOLRIS land cover categories (Fig. 1) for “impervious” (residential, industrial, commercial, and civic areas), “roads” (highways and roads), and “undifferentiated” (not mapped by the previous classes). We created a single land cover category for “greenspaces”, consisting of marsh lands, treed swamps, forests, and urban recreational lands (i.e., golf courses, and parks), and a category for “agriculture”, which represented the sum of tilled land and cultivated tree plantations. Undifferentiated areas were comprised of variable land cover types ranging from agricultural features to transportation rights-of-ways; therefore, any potential relationship with this land cover could not be readily interpreted. Thus, we removed all points that fell within this category (n = 27,469 or 3% of all points, n = 901,157). We also removed points that fell within water because this habitat type was only present in the buffer of three cats which made up <0.5% (n = 1,772) of the available land and was not used by any of the cats.

Statistical analysis

Home range size

To evaluate factors that could influence the size of a cat’s home range, we used generalized linear models (GLMs) with a gamma distribution and log link function. We performed separate GLMs for the 100% MCP and 95% KDE home ranges. Predictor variables for both the GLMs included intrinsic effects (age, sex, and neuroticism), and extrinsic effects (road density, the presence [1] or absence [0] of a major road, and time which was the ordinal date [Jan 1 = 1] of the first day of GPS data collection). Prior to model fitting, we first examined for signs of multicollinearity between predictor variables with a correlation matrix (Article S4, Fig. S2) conducted in R with the “corrplot” package (Wei & Simko, 2021). Model fit was evaluated by the second-order Akaike’s information criterion, corrected for small sample sizes (AICc; Burnham & Anderson, 2002). To avoid overfitting (Burnham & Anderson, 2002; Babyak, 2004), we used a two-stage model fitting procedure, by first fitting intrinsic predictors and then extrinsic predictors. For the intrinsic predictor model, we examined all possible combinations of intrinsic predictor variables (Article S4, Appendix S3). We defined influential variables as those that were retained in models within ΔAICc <2 (Burnham & Anderson, 2002). In addition, we further assessed variable support by examining if confidence intervals overlapped with zero (Arnold, 2010). If a variable was selected, it would then be used in the subsequent analysis that examined all possible combinations of extrinsic predictor variables (Article S4, Appendix S3). We used confidence intervals of 85% because when using AIC-based model selection, 95% confidence intervals may discard variables supported by a lower AIC value (Arnold, 2010). We removed 13 cats from the analyses due to missing variables (none had personality scores and the sex for nine were unknown) resulting in a sample size of 29 cats. For analyses that did not include sex or personality as co-variates, all 42 cats were included, except for the habitat analysis within a buffer, which included 41 cats. All GLMs were evaluated in R using the “MuMIn” package (Bartoń, 2022).

Habitat selection

To determine if cats showed a selection for land cover types, we used resource selection functions (RSF) to compare the land covers cats used with the land covers available to them (Manly et al., 2002). While second-order selection represents an animal’s selection of home range within a geographical area, and third-order selection describes the space within the home range, selection functions can be used to examine habitat availability at scales that fit between these orders (Northrup et al., 2021). We performed two RSF analyses at the third-order to compensate for a cat’s habitat selection being influenced by their owner. Therefore, we compared land cover use in a boundary around each owner’s house and within the home range (Johnson, 1980; van Moorter et al., 2016), detailed below.

We derived both RSFs by comparing the “used” GPS points to “available” points that were randomly generated anywhere within the set boundary at a ratio of 10:1 (10 random points per 1 used point; (Fieberg et al., 2021). To evaluate land cover used close to the owner’s house, we sampled the used and available points within a circular radius of 300 m, centered around each owner’s house. We used a boundary size of 300 m to stay consistent with previous research (Pirie, Thomas & Fellowes, 2022), which examined cat habitat selection within a site equivalent to a recommended buffer zone size of ∼300 m around a natural area (328 m, Thomas, Baker & Fellowes, 2014; 360 m, Lilith, Calver & Garkaklis, 2008). We removed one cat from this analysis because we did not have the exact address of the owner. For the second analysis, we sampled used and available locations within each cat’s 100% MCP home range.

We estimated the likelihood that a habitat type was selected by an individual by using binomial generalized linear mixed models (GLMM) with a logit link function. We constructed separate GLMMs for each habitat association close to the owner’s house and within the entire home range. In both analyses, we included a binary response variable (1 = used, 0 = available) to indicate if the cat used the land cover type or if it was available, where predictor variables included each land cover type at the point and whether it was used or not or (1 = within the habitat type, 0 = outside of the habitat type). To account for multiple locations occurring for individuals, cat ID was used as a random effect. The selection coefficient (β) from the RSF model indicates selection strength quantitatively for each land cover type with a positive or negative value reflecting a selection or not (Boyce et al., 2016; Avgar et al., 2017), respectively, and quantitatively with the β approaching 0 indicating a weaker selection (Avgar et al., 2017).

Results

Home range size

Our analysis of 42 individuals captured a high degree of variation in both the 95% KDE and 100% MCP home ranges (Fig. 3; Article S4, Appendix S4). The median 100% MCP was 4.40 ha with a range of 0.34–38.45 ha. While individual 95% KDE home range sizes were ∼4 times smaller, they showed similar variation between individuals (Fig. 4; Article S4, Appendix S4). The male and female 100% MCP home ranges had similar variation (Brown-Forsythe test, F = 0.28, p = 0.60; Fig. 4), with the median male 100% MCP home range of 6.15 ha (range: 0.34–38.45 ha) and a median female 100% MCP home range of 4.40 ha (range: 0.52–30.16 ha). We did not find evidence to suggest cats had larger home ranges at night than during the day (no support for hypothesis 2; Wilcox signed rank test, W = 410, p = 0.61). During the day, the median 100% MCP was 2.63 ha (range: 0.15–34.45 ha), while, at the night, the median 100% MCP was 3.63 ha (range: 0.14–22.88 ha; Fig. 5). The diurnal and nocturnal 95% KDE home range size showed similar patterns (Fig. 5; Article S4, Appendix S4).

Figure 3 Examples of owned cat home ranges tracked via GPS.

The 100% minimum convex polygon (translucent yellow) home ranges surround the 95% kernel density estimate (green) and 50% kernel density estimate (blue) home ranges.

Figure 4 A raincloud plot illustrating the data distribution of 100% MCP and 95% KDE home range data for male and female cats.

(A) 100% minimum convex polygon (MCP) home range sizes (ha) for males (blue) and females (pink) and (B) 95% kernel density estimate (95% KDE). The horizontal line within each boxplot represents the median value, while the upper and lower quartiles are represented by the upper and lower horizontal lines, respectively.

Figure 5 A violin plot illustrating the difference between each individual’s estimated diurnal and nocturnal 100% MCP and 95% KDE home range size.

The distribution of 95% kernel density estimate (95% KDE) and 100% minimum convex polygon (MCP) home ranges (ha) during both the day (yellow) and night (blue). The lines are used to connect each cat’s diurnal and nocturnal home range.

With respect to intrinsic predictor variables from 29 cats, the top ranked model was the null model and the second-ranked model (ΔAICc = 0.144) included age. Because the confidence intervals for sex (β = −0.24, 85% CI: −0.84, 0.37), age (β = 0.05, 85% CI: −0.02, 0.13; Table 1), and neuroticism (β = 0.002, 85% CI: −0.01, 0.02) did overlap with zero (no support for hypotheses 1, 3, or 5), we did not carry forward these predictors in the subsequent evaluation of extrinsic models. When considering which extrinsic variables could influence the size of the home range, there were four candidate models within 2 ΔAICc of the top model (Table 2). Based on the model averaged coefficient estimates and confidence intervals, 100% MCP home range size was positively related to road density (Table 3), suggesting traffic did not act as a barrier for cats (no support for hypothesis 4). The analysis of 95% KDE followed similar patterns to the 100% MCP models, but home range size was positively related to both road density and time of year (Article S4, Appendix S4). Data are available on Figshare: https://doi.org/10.6084/m9.figshare.25326322.v2.

Table 1 Model results of the top candidate GLM predicting which intrinsic factors influenced 100% MCP home range size of cats.

Age was measured in years.

Variable	Estimate	Lower 85% CI	Upper 85% CI	
Intercept	1.6756	1.0802	2.3225	
Age	0.0526	−0.0246	0.1314	

Table 2 Model selection results showing the top candidate models predicting what extrinsic factors could influence the cat’s 100% MCP home range size.

Road density was estimated by summing the road lengths, measured in meters, within a fixed boundary centred on each cat’s mean latitude and longitude coordinates. The variable “major road” indicated the presence or absence of a major road near the cat’s home range (binary). Roads were labeled as “major” based on Google Maps’ classification, related to traffic rates, and through “ground-truthing”.

Model	df	AICc	ΔAICc	Weight	
Road density	3	181.9789	0	0.2726	
Null	2	182.7196	0.7407	0.1882	
Major road + road density	4	182.8921	0.9132	0.1727	
Major road	3	183.4459	1.4670	0.1309	

Table 3 Model-averaged results for the candidate GLMs predicting factors that influenced the cat’s 100% MCP home range size.

Road density was estimated by summing the road lengths, measured in meters, within a fixed boundary centred on each cat’s mean latitude and longitude coordinates. The variable “major road” indicated the presence or absence of a major road near the cat’s home range (binary). Roads were labeled as “major” based on Google Maps’ classification, related to traffic rates, and through “ground-truthing”.

Variable	Coefficient	Lower 85% CI	Upper 85% CI	
Road density	0.0048	0.0001	0.0096	
Major roads	−1.0342	−2.0278	0.3508	

Habitat selection

Based on the random points generated within individual 100% MCP home ranges from 42 cats, the most abundant available land cover types were impervious surfaces (66%), roads (26%), greenspaces (6%), and least abundant was agricultural land (2%). Within the 100% MCP home ranges, greenspaces were predominately comprised of recreational parks (89%), and then various natural habitats (thicket swamps (4%), treed swamps (3%), deciduous forests (3%), and marshlands (1%)). Results from the GLMM provided some support for hypotheses 6 and 7, as cats selected to use impervious surfaces (73% of used locations) but avoided roads (22% of used locations) and greenspaces (4% of used locations), in proportion to their availability (Table 4, Fig. 6). There was also evidence that they avoided agricultural land (1% of land used; Table 4, Fig. 6). Although cats showed an avoidance of roads, used locations showed that 90% (n = 38) of cats utilized a road at some point during the study. The model results for habitat selection for a buffer close to the owner’s home were similar to the 100% MCP results (Fig. 6; Article S4, Appendix S5). Habitat data are available on Figshare: https://doi.org/10.6084/m9.figshare.25154555.v2.

Table 4 The selection coefficients (β) from a binomial logistic regression model for habitat selection within the 100% MCP home range.

Habitat types were obtained from the Southern Ontario Land Resource Information System (SOLRIS) v.3 (Ontario Ministry of Natural Resources and Forestry, 2019b).

Habitat type	Coefficient	Lower 95% CI	Upper 95% CI	p	
Impervious	0.36	0.34	0.38	<0.0001	
Greenspaces	−0.53	−0.58	−0.47	<0.0001	
Roads	−0.22	−0.25	−0.20	<0.0001	
Agricultural	−1.49	−1.68	−1.32	<0.0001	

Figure 6 Proportions of habitats used and available to cats within their 100% MCP home range and a 300 m boundary centered around their owner’s residence.

The proportion of habitat types used (purple) by cats based on individual GPS points and available (green) habitat types based on randomly generated points (created at a 10:1 random:used ratio) within (A) the 100% minimum convex polygon home range, and (B) a 300 m boundary centered around the owner’s house. Habitat types were obtained from the Southern Ontario Land Resource Information System (SOLRIS) v.3 (Ontario Ministry of Natural Resources and Forestry, 2019b) where “impervious” includes residential, industrial, commercial, and civic areas, “road” includes highways and all other roads, “greenspaces” includes recreational parks and various types of natural land cover including marsh lands, swamps, and mixed forests, and “agricultural” includes tilled land and cultivated tree plantations.

Discussion

Our study demonstrated that home range sizes of owned cats in the Kitchener-Waterloo-Guelph-Cambridge region of southwestern Ontario were highly variable among individuals. The largest 100% MCP home range size we estimated was 38.45 ha, which was more than 100x larger than the smallest 100% MCP home range size (0.34 ha). Previous studies on owned cats have reported a similar minimum 100% MCP home range size (i.e., ≤ 1 ha) but maximum home range sizes across studies vary considerably (Barratt, 1997; Meek, 2003; Morgan et al., 2009; Mesters, Seddon & van Heezik, 2010; van Heezik et al., 2010; Thomas, Baker & Fellowes, 2014; Fardell et al., 2021). However, comparisons between studies should be approached with caution due to differences in geographic location, predator communities, study duration, modelling techniques, and technology. For example, studies that used hand-held radiotelemetry reported differences of 10 ha (Morgan et al., 2009) and 15 ha (Meek, 2003) between the maximum and minimum home range size, while studies using GPS tracking reported differences of 20 ha or more (Mesters, Seddon & van Heezik, 2010; van Heezik et al., 2010; Thomas, Baker & Fellowes, 2014). Locations of cats taken via radiotelemetry are typically less frequent (daily, weekly: Barratt, 1997; Meek, 2003; Morgan et al., 2009) compared to automatically downloaded GPS points taken every minute or 15 min (Mesters, Seddon & van Heezik, 2010; van Heezik et al., 2010; Thomas, Baker & Fellowes, 2014), which means that the former method has a much lower probability of capturing infrequent forays cats make outside their primary areas of use compared to the latter. Furthermore, as we have demonstrated (Article S4, Fig. S1), a large number of points (>100) are needed to accurately capture 100% MCP home range size, something that radiotelemetry-based estimates rarely achieve.

Our finding that home range size was positively associated with road density suggests that roads may facilitate cat movement. However, according to our habitat selection results, cats used roads less often than expected. The combination of these results suggest that cats may not walk directly on the road but instead travel along the roadside, using embankments or vegetation and, thus, avoiding open areas (Barratt, 1997; Meek, 2003). Moreover, roads may dissuade the presence of other cats and predators, allowing for more overall movement and thus larger home ranges. Additionally, most home ranges did not include major roads, further suggesting road avoidance (Article S4, Fig. S6). Cats may not like walking next to major roads because the sidewalks have little vegetative cover and the noises from traffic could cause stress (Eagan, 2020).

The similarity in home range size between males and females is consistent with some previous studies (van Heezik et al., 2010; Hanmer, Thomas & Fellowes, 2017; White, 2019; Cecchetti et al., 2021) and could be explained by the fact that all cats were desexed. Breeding males in the Felidae family can roam far distances in pursuit of females (Liberg et al., 2000; Palomares et al., 2017), but desexed domestic males show similar hormone profiles (Hart & Hart, 2021) and traits as females (Liberg et al., 2000). On the other hand, studies involving large sample sizes ranging from 79 (Pirie, Thomas & Fellowes, 2022) to 875 (Kays et al., 2020) of predominately desexed, owned indoor-outdoor cats did find significant differences between male and female home ranges (Hall et al., 2016; Kikillus et al., 2017; Roetman et al., 2017; Kays et al., 2020), although see Cecchetti et al. (2021) and Jensen et al. (2022). One reason for this discrepancy could be an unmeasured factor which is the age at which a cat is neutered (Hall et al., 2016). Males may maintain a larger home range if they are neutered after sexual maturity rather than before because they already established their home range (Bradshaw, 1992; Hall et al., 2016), and some sexual behaviours may be maintained for periods of time even after neutering (Rosenblatt & Aronson, 1958; Lisk, 1967; Carter, 1997). However, the differences found between these studies may also reflect differences in sampling designs and ownership status (e.g., Guttilla & Stapp, 2010; Gehrt et al., 2013), as not all studies may have defined “owned” cats equally. Ultimately, comparisons in home range sizes between sexes require more attention because, if sex differences do exist, then already resource-intensive programs, such as trap-neuter-release (TNR), may find efficiencies by targeting a specific sex. For example, if males tend to roam more than females, then focusing neutering efforts on males might have a larger impact on reducing conflict with neighbours and lowering disease transmission rates (Loyd et al., 2013), compared to equal TNR effort across sexes.

Interestingly, we found some evidence that time of year influenced 95% KDE home range but not 100% MCP home range, suggesting that the size of a cat’s most heavily used areas were largest in the fall compared to the spring and summer, but not the total area used, which included longer forays outside the core area of use. While this is somewhat difficult to explain, one possibility is that, if the 95% KDE home range primarily represents space that is actively defended (i.e., territory; Burt, 1943) then such areas may expand over the year as neighboring cats are more likely to be kept indoors during the fall, lowering local density (Clyde et al., 2022). Points outside the 95% KDE home range, in contrast, may primarily represent off-territory forays that are either not dictated by the presence of conspecifics or are consistently areas never occupied by conspecifics. It is also possible that 95% KDE home range estimates are more sensitive to changes in prey availability than 100% MCP home range estimates because the former is a better representation of primary hunting areas. For example, if prey availability is lower in the fall than spring or summer, then this would result in a larger 95% KDE but no change in the 100% MCP home range. Testing this idea, however, would require collecting data on prey availability in conjunction with cat movements across multiple seasons. In addition to the non-trivial task of accurately sampling prey availability, we also note that it can be challenging to convince owners to commit to tracking their cats over longer timeframes (i.e., multiple seasons).

Our results provide evidence that cats selected for impervious surfaces (urban and residential areas), which is consistent with previous studies conducted in the same region (Flockhart, Norris & Coe, 2016; Clyde et al., 2022) and unsurprising given their close relationship to their owners who provide them with food and shelter. We also found that cats avoided both greenspaces and roads which provide support for the possibility that predators such as coyotes could deter cats from entering greenspaces (Gehrt et al., 2013; Kays et al., 2015) and that traffic may deter cats from using roads (Barratt, 1997). Although we did not collect data related to the presence or absence of coyotes, previous studies in urban areas, using GPS (Thompson, Malcolm & Patterson, 2021), radio telemetry (Quinn, 1997; Grubbs & Krausman, 2009; Gehrt et al., 2013), and trail cameras (Kays et al., 2015; Clyde et al., 2022), have reported that coyotes affected cats either directly from predation (Grubbs & Krausman, 2009; Gehrt et al., 2013) or indirectly through avoidance (Gehrt et al., 2013; Kays et al., 2015), including within our study region (Clyde et al., 2022).

Management Implications and Recommendations

Based on our findings, we provide four main recommendations for interest groups to consider when managing outdoor cats. First, despite a general avoidance of roads, 90% of cats in this study crossed roads or walked alongside them placing them at risk of blunt force trauma or death from vehicle collisions. Findings such as these may be valuable for interested parties such as veterinarians or shelter workers to present and discuss with cat owners to help them assess the risk for allowing their cat(s) to wander outside unsupervised. Second, consistent with Clyde et al. (2022), our finding of a close association with residential and urban land covers indicate that native wildlife species found close to buildings are at the greatest risk of predation, and may have implications for backyard bird feeders (Dunn & Tessaglia, 1994) or for species that can take advantage of the novel ecosystems created by humans (e.g., song sparrows (Melospiza melodia), blue jays (Cyanocitta cristata), and cedar waxwings (Bombycilla cedrorum)). Third, our results suggest that cat curfews are unlikely to reduce the spread of diseases carried by cats and the magnitude of risks that cats face while outside, as cats roamed the same distances during the two time periods we examined. Lastly, based on our habitat selection analysis, buffer zones around greenspaces may be of limited use for eliminating predation rates of wildlife by owned cats in urban habitats. That said, if the primary goal of a community is to prevent wildlife predation in greenspaces by cats, then buffer zones could be an option. To estimate a buffer zone based on our results, we added 20% safety margin (Lilith, Calver & Garkaklis, 2008) to the largest 100% MCP home range (38.45 ha, which is equivalent to a diameter of 700 m), resulting in an 840 m buffer. This conservative approach would encompass even occasional cat forays into natural habitats. Regardless, results of actual predation events from the Catcams (the cameras attached alongside the GPS units) that will arise from this study should provide a better understanding of where and when predation is greatest, information that will complement data on cat habitat use; therefore, providing a more comprehensive picture of predation risk across the urban landscape.

Our findings highlight that the movement of outdoor domestic cats is highly variable among individuals. This finding is consistent with other studies that have shown high individual variation in hunting behaviour (Morgan et al., 2009; Tschanz et al., 2011; Dickman & Newsome, 2015; Cecchetti, Crowley & McDonald, 2020) and risk-taking behaviours (Loyd et al., 2013; Bruce et al., 2019), making it hard to develop a one-size-fits-all approach to cat management. While a buffer size of 840 m may protect native wildlife from cat depredation in our study area, this could be an overestimate (Lilith, Calver & Garkaklis, 2008; Thomas, Baker & Fellowes, 2014; Hanmer, Thomas & Fellowes, 2017) or underestimate (Mesters, Seddon & van Heezik, 2010) in other locations. Given the complexity of urban environments and our findings showing how cat movement was related to surrounding habitat, it is hard to develop a one-size fits all buffer size that would be appropriate in all cities. Additional research on cat movement may help reduce some of these complexities and provide a way to better assess how to adjust buffer size based on surrounding habitat, but first it is valuable to consider if such research is necessary to meet cat management or wildlife conservation objectives. For municipalities or non-governmental organizations that are considering implementing management strategies that incorporate buffers they would likely benefit from conducting locally or regionally based research to collect relevant movement data from local cats. This could be done with a relatively small sample size of 20–30 individuals because, as our research has shown, this is enough to capture variation in home range sizes.

Based on the precautionary principle, which advocates for taking preventative action when there is uncertainty (Kriebel et al., 2001), and the high variability of cat movement, restricting cats indoors would be the main way to fully reduce their environmental impacts. However, this would ignore the complexity of issues surrounding outdoor domestic cats and society’s perceptions of cats, such that this management strategy would be impractical and likely infeasible. For example, the primary reason owners keep their cats inside is the concern for their cat’s safety or wellbeing (MacDonald, Milfont & Gavin, 2015; Tan et al., 2021; van Eeden et al., 2021) and not because cats hunt wild animals (Wald, Jacobson & Levy, 2013; van Eeden et al., 2021), suggesting that focusing on environmental impacts is not an effective communication strategy (Gramza et al., 2016; Crowley, Cecchetti & McDonald, 2020b; van Eeden et al., 2021). We encourage future research to continue to broaden understanding how cats behave outside to develop the best management practices and provide stakeholders such as shelter workers and veterinarians with information that can help owners make informed decisions about their cat’s outdoor roaming activities.

Supplemental Information

Supplemental Information 1 Qualifying survey

Supplemental Information 2 Code for calculating the distance and speed between GPS points

Supplemental Information 3 Supplementary materials

Supplemental Information 4 Cat personality survey

Supplemental Information 5 Author checklist

We would like to thank Hannah Clyde and Karl Heide for collecting field data in 2019. We also thank Kat Albrecht-Thiessen for discussions surrounding outdoor cats and their personality. Finally, we thank Andie Siemens, our other field technicians, and cat owners for their hard work and dedication to this project.

Additional Information and Declarations

Competing Interests

Author Contributions

Animal Ethics

Field Study Permissions

Data Availability

The authors declare there are no competing interests.

Marlee L. Pyott performed the experiments, analyzed the data, prepared figures and/or tables, authored or reviewed drafts of the article, and approved the final draft.

D. Ryan Norris conceived and designed the experiments, analyzed the data, prepared figures and/or tables, authored or reviewed drafts of the article, and approved the final draft.

Greg W. Mitchell conceived and designed the experiments, analyzed the data, authored or reviewed drafts of the article, and approved the final draft.

Leonardo Custode analyzed the data, prepared figures and/or tables, and approved the final draft.

Elizabeth A. Gow conceived and designed the experiments, performed the experiments, analyzed the data, prepared figures and/or tables, authored or reviewed drafts of the article, and approved the final draft.

The following information was supplied relating to ethical approvals (i.e., approving body and any reference numbers):

All methods were approved by the University of Guelph’s Research Ethics Board (approval #4189), and the University of Guelph’s Animal Care Committee (AUP #4183).

The following information was supplied relating to field study approvals (i.e., approving body and any reference numbers):

University of Guelph’s Research Ethics Board

The following information was supplied regarding data availability:

Cat characteristic data are available at Figshare: Pyott, Marlee; Norris, Ryan; Mitchell, Greg; Custode, Leonardo; Gow, Elizabeth (2024). Data collected from each cat in the sample. figshare. Dataset. https://doi.org/10.6084/m9.figshare.25326322.v2.

The habitat selection data are available at Figshare: Pyott, Marlee; Norris, Ryan; Mitchell, Greg; Custode, Leonardo; Gow, Elizabeth (2024). Habitat selection data. figshare. Dataset. https://doi.org/10.6084/m9.figshare.25154555.v2.

The code is available in the Supplementary File.

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
