# Peer review of "Home range size and habitat selection of owned outdoor domestic cats (Felis catus) in urban southwestern Ontario"

_PeerJ, doi:10.7717/peerj.17159_

## Round 0.1 · original submission · Minor Revisions

Two expert reviewers have provided reasonably positive assessments of your work. However, each indicate unique points in need of clarification. Reviewer 2 has provided extensive suggestions and comments in an annotated PDF. Please be sure to review all suggested changes carefully and provide a response or rebuttal along with your revision. I found your paper to be clear and well-written and to address an important and timely topic. I applaud your use of citizen science. I have only a few minor points of my own for you to consider or correct:
Line 137 is missing a “to” between “lead” and “higher.”
Line 146, “males” should be “male.” Should this refer to intact cats only? If none of your cats were intact, perhaps this hypothesis does not apply?
Line 152, do you mean “more active than..?”
Line 235, should be improbable
Line 274, delete “by.”
You should report inter-item reliability for items assessing the same personality trait.
Please report exact p values and confidence intervals where applicable.

Reviewer 1 ·

Basic reporting

I find this research to be very well communicated with clear language throughout. The introduction expounds well on the relevant background information and the goals and hypotheses are stated clearly at the end of the introduction. The manuscript follows standard structure and the methods, results, and discussion are all easy to follow and line up with the stated goals of the paper.
The tables and figure are excellent and help to communicate the project; I just have a few suggestions on them:
• For Table 1 I recommend renaming the ‘intercept’ model as the ‘null’ model or something similar, it is confusing as-is and the term ‘null model’ is used in text (though in reference to a different model).
• I would suggest clearly identifying the units of each measure in the tables, especially for Table 2. This would help with interpreting the models, especially the coefficients in Table 2 so readers know what the measures are for road density & major roads, and what outcome variable they are influencing.
• The mapping colors in Figure 1 are a little washed out and hard to distinguish for my liking; it would be nice to use more distinct colors for the greenspaces, agricultural land, and undifferentiated land. The inset map in the bottom right could also be improved- I recommend removing most of the background lines and blurred city names, making the main boundaries (such as between land & water, or major roads) darker and clearer, and exaggerating the lines and labels for the study area.
• I think it would be useful to include more of the models in the model selection tables, to show how much poorer other models that included other parameters of interest were. The whole model competition should be included as supplementary material as well.
• Figure 4 shows little difference in male and female range size averages, but it might be worth examining if male and female cats have different variance in their home range sizes. This is not essential but the figure begs this question, and there is much work showing greater male variability in behavior across mammals.

Experimental design

This study used an observational design- cats were recruited, GPS collars were used to track the cats for specified amount of time, and then data related to the cats and their space use was analyzed. The protocols followed are explained in detail and the use of the data after data collection is well explained. I think this study’s methods allow it examine its primary questions and goals well.
My only question/potential suggestion is clarification on why the authors’ analyzed habitat selection via the individual data points rather than using the calculated home ranges. The way the paper was set up I thought the author’s might compare the % of MCP and KDE that are different habitat types to the % of these habitat types in the surrounding area. I do not see anything flawed with the approach that was taken, but it might be worth adding a sentence justifying the chosen analytic approach given that home ranges were already calculated.

Validity of the findings

The findings are presented clearly and the figures are used well to visually re-inforce the main findings. Based on the sound methodology and clear explanations I find the results valid with nor problems. I have a few suggestions though:
• It would be useful if all of the main hypotheses mentioned in the introduction were at least referenced in the results, if only to say that model selection indicated home range size was not affected by sex, etc.
• L483-490- perhaps it is food-related as well? I.e. less hunting options means cats expand their main roaming area.
• L527-528- is this claim (larger home ranges = more hunting) backed up by the cited paper? I could not tell.
• I do not see all of the possible data and R code used for the analyses. The R code provided uses shapefiles, and this data is not provided. The data that breaks down home ranges by day & night, and the habitat available vs. used data, are not provided. I understand data that shows the actual GPS points or actual home range locations are not provided due to confidentiality reasons. However, I think the data I mentioned could be made available.

Additional comments

My only general comment is that, given the low use of greenspaces by cats, the ‘Management Implications and Recommendations’ might warrant some minor revision. When I see that cats are actually avoiding greenspaces overall this might suggest that management interventions for owned cats are of only limited utility because these cats are rarely using habitats of concern as it is. Perhaps the author could address this, by moderating their recommendations or addressing why these findings.

Reviewer 2 ·

Basic reporting

The issue of outdoor cat ecology and management is presented in a suitable fashion with a fairly complete presentation of references. At times the writing is perhaps a little too casual in places, and there are places where clarity is limited such that it would be difficult to replicate this study. I provide examples in specific comments attached.

One of the complexities of outdoor cat research is the degree of ownership involved, and the amount of time they spend outside. In this case, ownership is certain and well documented, but I got confused in the study description as to how much time cats were outside, and I’m not sure the authors really distinguish the activity/space use of the indoor-outdoor cats versus unowned free-ranging cats. Other work, mainly with cameras, suggest a difference (Bennett et al. 2021).

Some editing is needed for clarity, accuracy, and grammar. Specific comments attached, but note that I did not do a thorough editorial review.

The tables and figures are generally clear and organized. I recommend putting 100% MCP in each for clarity.

I appreciated the information provided in the supplemental files. I can understand why the authors could not provide more spatial information regarding cat locations.

Experimental design

The research question and subsequent objectives are well defined. My impression after reading the ms was that the design was at times somewhat loosely structured to the detriment of the study. Examples include the lack of basic information on the subject animals in the first year, and even some information on cats in the second year. We don’t know how long cats were monitored or why they weren’t monitored during different seasons. However, it is perhaps to be expected to some degree when working with citizen scientists and the complexities of trying to study indoor-outdoor, owned cats. See attached for more comments regarding my questions about the design.

The analytical approach appeared to be appropriate, although a minor question is why use 100% MCP if part of the goal is to compare with other studies, which almost always use 95%. Modeling strategies with intrinsic and extrinsic factors appeared to be sound.

Validity of the findings

The subject matter is timely and would be of interest to a wide readership. The issue of free-ranging cat management, and the ecological impacts of cats, is an international conundrum. The authors present some interesting data using relatively new GPS trackers. The strengths of the paper are the use of GPS trackers for high resolution spatial data, an appropriate analytical approach, and they provide a practical application of the results. See my attached comments for more details about where I think improvements are needed.

Additional comments

I provide many comments on the attached file.

Annotated reviews are not available for download in order to protect the identity of reviewers who chose to remain anonymous.

---

## Round 0.2 · Minor Revisions

I appreciate your thorough revision and detailed response letter. I invited the previous reviewer who recommended major revisions to re-review your paper, but that reviewer has not responded to the request so, in light of the overall positive view of your work by myself and both of the previous reviewers, I have decided to proceed with my own read of the paper rather than introducing a new reviewer at this point.

Previously, I requested that you report inter-item reliability for your personality measure. Although reliability for an initial sample may be available to show the measure has been validated, authors must still provide reliability data for their specific sample whenever multiple-item responses are being used to infer an underlying trait.

Lines 308-311 I do not think that it makes sense to describe this as calculating the percentage of neuroticism. Just say you reported the average score and standardize it. If you report descriptives, it will be most intuitive if the scores retain their scored values of 1-7.

It is my understanding that FeLV cannot be transferred to other animals, including humans (check lines 52-53 please).

Line 128, place a , after cats

Line 175 delete the “d” on “ranged”

Lines 182-183, I might change “due to low cat owner motivation and willingness” to sound less negative toward the citizen scientists to something like “to increase the likelihood of compliance with the protocol” or something like that.

Line 201, 276, 318, replace “since” with “Because.” Avoid using since and while in non-temporal contexts.

Place , after e.g. and i.e. throughout (e.g., line 206).

Line 207, should “out” be “outside” or “outdoors?”

Is it not possible to contact the owners who participated in the study in Year 1 to obtain information about the sex of their cats etc.?

Line 339, place ; before “therefore” and , after. Place all punctuation within quotations throughout.

Is the word data missing on line 353?

Line 510, is “as” missing after “such?”

Line 516, change cats’ to cat’s

---

## Round 0.3 · accepted · Accept

Thank you for making these final minor corrections so quickly!